# Evaluating Fermentation Quality, Aerobic Stability, and Rumen-Degradation (In Situ) Characteristics of Various Protein-Based Total Mixed Rations

**DOI:** 10.3390/ani13172730

**Published:** 2023-08-28

**Authors:** Halidai Rehemujiang, Hassan Ali Yusuf, Tao Ma, Qiyu Diao, Luxin Kong, Lingyun Kang, Yan Tu

**Affiliations:** 1Key Laboratory of Feed Biotechnology of Ministry of Agriculture and Rural Affairs, Institute of Feed Research of Chinese Academy of Agricultural Sciences, Beijing 100081, China; halidai@caas.cn (H.R.); hassanagri99@gmail.com (H.A.Y.); matao@caas.cn (T.M.); diaoqiyu@caas.cn (Q.D.); kongluxin@caas.cn (L.K.); 18821622269@163.com (L.K.); 2Faculty of Veterinary Medicine and Animal Husbandry, Somali National University, Mogadishu P.O. Box 15, Somalia

**Keywords:** aerobic stability, cottonseed meal, fermentation, rapeseed meal, total mixed ration, rumen digestibility

## Abstract

**Simple Summary:**

Fermented total mixed ration (FTMR) is a method used to ferment feed under anaerobic conditions in a tightly sealed container, with the potential to enhance nutrient utilization and extend the shelf life of feed. We found that the loss of aerobic stability may be associated with the dominant yeast species found in FTMR; however, there is no discernible relationship between yeast counts and the loss of aerobic stability. However, the replacement of soybean meal (SBM) with cottonseed meal (CSM) and rapeseed meal (RSM) in total mixed rations (TMRs) by fermented TMR has an effect on the degradation of anti-nutritional factors. In order to explore the effects of fermentation and aerobic exposure to FTMR on nutrients and anti-nutritional factors, this study aimed to ascertain the impact of *B. clausii* and *S. cariocanus* inoculations on TMR fermentation quality, aerobic stability, anti-nutritional factors, and in situ rumen-degradation characteristics’ variables. We found that TMRs fermented with inoculations of *B. clausii* and *S. cariocanus* improved the fermentation quality and nutrient composition, decreased the anti-nutritional factor content, and, thus, achieved detoxification. Meanwhile, the effective disappearance of nutrients in the rumen was increased.

**Abstract:**

The purpose of this experiment was to evaluate changes in fermentation quality, chemical composition, aerobic stability, anti-nutritional factors, and in situ disappearance characteristics of various protein-based total mixed rations. Soybean meal (control, non-fermented), fermented cottonseed meal (F-CSM), and fermented rapeseed meal (F-RSM) group were used to prepare the TMRs with corn, whole-plant corn silage, corn stalks, wheat bran, and premix. The test groups were inoculated at 50% moisture with *Bacillus clausii* and *Saccharomyces cariocanus* and stored aerobically for 60 h. The nylon-bag method was used to measure and study the rumen’s nutrient degradation. The pH of all TMRs after 48 h of air exposure was below 4.8, whereas that of the F-CSM and control and F-RSM groups increased to 5.0 and >7.0, respectively. After 8 h of aerobic exposure, the temperatures of all groups significantly increased, and 56 h later, they were 2 °C higher than the surrounding air. The lactic acid concentration in the F-CSM and F-RSM groups increased after 12 h of aerobic exposure and then decreased. The acetic acid concentrations in the fermented groups decreased significantly with the increasing air-exposure time. The yeast population of the TMRs increased to more than 8.0 log_10_ CFU/g before 72 h of air exposure, followed by a decrease in the population (5.0 log_10_ CFU/g). After fermentation, the free gossypol (FG) concentration in F-CSM decreased by half and did not change significantly during the air-exposure period. Fermentation with probiotics also reduced the F-RSM’s glucosinolate concentration, resulting in a more than 50% detoxification rate. Compared with the F-CSM and F-RSM groups, the effective degradation rates of nutrients in the control group were the lowest, and the dry matter (DM), crude protein (CP), natural detergent fiber (NDF), and acid detergent fiber (ADF) all degraded effectively at rates of 28.4%, 34.5%, 27.8%, and 22.8%, respectively. Fermentation with *B. clausii* and *S. cariocanus* could improve the fermentation quality and nutrient composition, decrease the anti-nutritional factor, and increase nutrient degradation of the TMR with cottonseed meal or rapeseed meal as the main protein source, thus achieving detoxification.

## 1. Introduction

Proteins are an indispensable nutrient source for animal growth and development. The quality of protein feed affects the health and productivity of animals. Soybean meal is a high-quality protein feed, but its high price has seriously affected its use in the domestic-animal-breeding industry, especially in China. 

In China, the annual output of mixed-meal feed, such as cottonseed meal, is more than 30 million tons. However, animal nutritionists have increasingly focused on plant protein substitutions for soybean meal (SBM). Due to their lower prices and high crude CP content, cottonseed meal (CSM) and rapeseed meal (RSM) are of interest. CSM, a by-product of cottonseed oil extraction, has a CP content of 34–40%, a crude fiber (CF) content of 11%, a neutral detergent fiber content of 25–30%, and comparatively high levels of organic phosphorus and vitamin B [1]. Rapeseed is crushed to produce RSM after the oil has been extracted. It has a high protein content (34–38%), an amino acid composition that is well balanced, and an NDF content of 25–30% [2]. The RSM protein contains more sulfur amino acids than other plant proteins and has a nutritional value comparable to that of the SBM protein. However, because they include anti-nutritional components such as erucic acid and glucosinolate in RSM and FG in CSM, their use in animal diets is still limited [3].

Numerous detoxification strategies have been developed for CSM and RSM, including biological [4], chemical [5], and physical [6,7] treatments. Yeasts and *Aspergillus oryzae* are the most commonly used microorganisms of solid fermentation [8]. Physical detoxification methods include heating the rapeseed cake by using dry heat and steam. This method of operation is simple and inexpensive, but the detoxification rate is low and irregular [9]. Chemical detoxification involves the addition of chemicals to the rapeseed. Under certain conditions, glucosinolates in rapeseed undergo a glycosidase reaction and develop poisons. The method is simple and highly targeted; however, owing to the effect of a single seedling, blending makes it difficult to achieve the effect [10].

Our previous study confirmed that the fermentation with 1.0 × 10^9^ and 5.0 × 10^9^ CFU/kg DM for *B. clausii* and *S. cariocanus* to TMR with CSM and 1.0 × 10^10^ and 5.0 × 10^9^ CFU/kg DM for *B. clausii* and *S. cariocanus* to TMR with RSM improved the nutritional value and decreased the content of anti-nutritional factors [11]. In addition, an increased CP content and decreased NDF content, as well as a reduction in the content of anti-nutritional factors in both TMRs containing CSM/RSM, were observed after fermentation. Therefore, the following were the ideal fermentation conditions for fermented TMR (FTMR) with CSM: mixed microbial strains (1.0 × 10^9^ and 5.0 × 10^9^ CFU/kg DM for *B. clausii* and *S. cariocanus*, respectively) incubated aerobically for 60 h at 32 °C and 50% humidity. The following were the ideal fermentation conditions for FTMR incorporating RSM: 60 h incubation, 28 °C temperature, 50% moisture, and mixed microbial strains (1.0 × 10^10^ and 5.0 × 10^9^ CFU/kg DM for *B. clausii* and *S. cariocanus*, respectively) [12]. These findings served as the basis for this study’s fermentation conditions.

Fermented TMR (FTMR) refers to a type of ruminant feeding technology in which the finished TMR is inoculated by a particular compound strain [13] for anaerobic fermentation so that it can be stored for a predetermined amount of time. In comparison to TMR, FTMR preparation is more efficient and less time-consuming for daily feeding management, so it may significantly save labor and resources [14]. In the lack of TMR mechanical equipment, FTMR can be purchased and used by small-scale farmers, and it can be transported over long distances [14]. Moreover, FTMR has the potential to enhance feeding levels while also promoting the economic development of animal husbandry [15]. A TMR optimization method using *L. casei TH14* and fermented sugarcane bagasse significantly changed the intake, digestibility, rumen ecology, and milk production of mid-lactation Holstein cows [15].

The profitability of farms is severely affected by the aerobic decomposition of silage. In addition to being less palatable, spoiled silage also has a negative impact on livestock productivity. FTMR results in highly enhanced aerobic stability [16,17]. This makes it possible to transport and use FTMR in various ways for a considerable amount of time, without experiencing aerobic deterioration [16]. Numerous studies have investigated the aerobic stability of FTMR. Yeast counts can decrease below the limit of detection (10^2^ colony forming units CFU/g) when anaerobic fermentation is sustained for a month or more, substantially enhancing aerobic stability [18]. Silage with a high yeast population (>10^5^ CFU/g) has been observed to spoil upon contact with oxygen [19]. Even with more than 10^6^ CFU/g of yeast measured at silo opening, aerobic stability in FTMR was attained [13]. This shows that the loss of aerobic stability may be associated with the dominant yeast species found in FTMR; however, there is no discernible relationship between yeast counts and the loss of aerobic stability. However, the replacement of SBM by CSM and RSM in TMR by fermentation has an effect on the degradation of anti-nutritional factors. To explore the effects of fermentation and aerobic exposure to FTMR on nutrients and anti-nutritional factors, this study aimed to ascertain the impact of *B. clausii* and *S. cariocanus* inoculations on TMR fermentation quality, aerobic stability, anti-nutritional factors, and in situ rumen-degradation characteristics’ variables.

## 2. Materials and Methods

### 2.1. Preparation of a Total Mixed Ration and Anaerobic Fermentation

The entire corn plant after cob harvest at the wax-ripe stage (Sanbei 21) was cut immediately when part of the leaves remained green after ear picking on 30 August 2021, at the wax-ripe stage. The following ingredients were used to manufacture TMR: whole-plant corn silage (lactic acid at 67.63 g/kg DM, acetic acid at 15.88 g/kg DM, and propionic acid at 1.72 g/kg DM), corn stalk, SBM, CSM, RSM, wheat bran and fat powder, urea, and premix. The dietary feed was formulated according to the guidelines for meeting the nutrient requirements of sheep and allowing them to gain 300 g per day (NRC, 2007). The diet’s components and chemical composition are listed in Table 1.

A mixture of the microbial strains at a ratio of 1:5 (1.0 × 10^9^ CFU/kg DM *B. clausii:* 5.0 × 10^9^ CFU/kg DM *S. cariocanus*) was inoculated into the FTMR with CSM (F-CSM) at 50% moisture content. A moisture content of 50% and a microbial strain combination ratio of 1:5 (1.0 × 10^10^ CFU/kg DM *B. clausii:* 5.0 × 10^9^ CFU/kg DM *S. cariocanus*) were used for the FTMR with RSM experiment. The combination was fermented for 60 h at 32 °C for F-CSM or 28 °C for F-RSM in a fermenter machine (Model SSJX-WH-3.0, Shengshun Machinery Manufacturing Co., Ltd., Shenyang, China) with a capacity of 500 kg volume. The mixture was combined with silage after fermentation and mixed before being stored in a plastic bag (using the V5 Reelanx vacuum sealer). The TMR in the control group was mixed when the other two groups finished fermentation. Except for the non-fermented TMR, F-CSM, and F-RSM groups during 60 h anaerobic fermentation, when they were finished going through the fermentation process, fermented groups were tested for aerobic stability. Three TMRs were used in this study.

### 2.2. Aerobic Stability Tests

Three 6 L polyethylene barrels were set up in triplicate for each TMR treatment, and 10 kg of the substance was introduced to each barrel, without compacting, and fully mixed for the aerobic stability test. The remaining contents were loosened, returned to the barrel, and left uncovered for the duration of the air-exposure period. Half of the contents (weight) were discarded. After the aerobic stability test began, sub-samples of non-fermented and FTMRs were collected at 0, 12, 24, 48, 72, 96, 120, 144, and 168 h. The temperatures of the ambience and the TMRs were measured with a mercury thermometer, and aerobic deterioration was deemed to have occurred when the temperature difference between the TMRs and the atmosphere exceeded 2 °C [13]. 

### 2.3. Fermentation Profiles and Microbial Counts 

About 20 g of sample was blended with 180 mL distilled water and macerated for 24 h at 4 °C. The extract was filtered through 4 layers of a gauze. The filtrate was used for pH, volatile fatty acid (VFA), and ammonia nitrogen (NH_3_-N) determinations. The pH was measured with a pH meter (Testo-206-pH, Testo Co., Berlin, Germany). VFA profiles were determined using a gas chromatography instrument (GC-6800; Beijing Beifen Tianpu Instrument Technology Co., Ltd., Beijing, China) [20]. The NH_3_-N was determined using the phenol-hypochlorite reaction method [21].

The microorganism populations were enumerated using the technique described by Xu et al. [22]. Samples (10 g) were serially diluted from 10^−1^ to 10^−9^ in a mixture of 90 mL of sterilized distilled water. At appropriate dilutions, colonies were enumerated from the plates, and the CFU was expressed per gram of dry matter (DM). After 48 h of anaerobic incubation at 37 °C, the presence of lactic acid bacteria (LAB) was determined by plate counting on deMan, Rogosa, and Sharpe agar (Difco Laboratories, Detroit, MI, USA). On nutritional agar medium (Nissui-Seiyaku Ltd., Tokyo, Japan) incubated for 48 h at 30 °C under aerobic conditions, aerobic bacteria were enumerated. After incubation for 24 h at 30 °C, yeasts were counted on potato dextrose agar (Nissui-Seiyaku Ltd., Japan) that was acidified with a sterilized tartaric acid solution to pH 3.5. Based on the appearance of the colony and the shape of the cells, yeast and aerobic bacteria were separated. By streaking each yeast colony on peptone–dextrose agar, each colony was purified. The purified strains were then stored at −80 °C with 10 g/L glycerine for investigations (Difco Laboratories, USA).

### 2.4. Chemical Composition and DM Loss

To determine the DM content, 800 g of the sample was dried at 65 °C for 72 h. The DM loss in the samples was calculated from the difference in weight before and after fermentation. Samples were ground using a Wiley Mill after drying (1 mm screen, Arthur H. Thomas, Philadelphia, PA, USA). We tested OM use GB/T 6435-2014 [23] and GB/T 6438-2007 [24], ether extract (EE) (Horwitz et al., 1970), and true protein (TP) by using Sniffen et al.’s (1992) method; and NDF (add 0.5 mL/sample Alpha Amylase, Ankom A2000 and A200 instrument, Macedon, OH, USA) and ADF by using an Ankom 200 system (Ankom Technology Corporation, Macedon, NY, USA) according to the manufacturer’s instructions. Starch was determined by the perchloric acid hydrolysis method [25]. Using the N data collected with a Foss KJELTEC 8400 analyzer, the CP was estimated as the multiplication of the total N by 6.25. (FOSS; Nil Foss, Hilleroed, Denmark). GE used a Parr-6400 oxygen and nitrogen calorimeter (Parr-6400, Beijing Oriental Shenglongda Technology Co., Ltd., Beijing, China) to determine the oxygen and nitrogen levels.

### 2.5. Anti-Nutritional Factor Analyze 

The method approved by the American Oil Chemists Society [26] was used to calculate the free gossypol concentration. Accordingly, the palladium chloride method was used to determine the F-RSM glucosinolate concentration [27].

### 2.6. In Situ Disappearance of Various TMR

A nylon cloth with a diameter of 300 mesh or 35~50 µm was cut into a rectangle measuring 17 × 13 cm; the rectangle was then folded in half, and we used nylon thread to sew it twice. A 12 × 8 cm nylon bag was prepared, with the edge of the bag sealed using a soldering iron or on the alcohol lamp to avoid silk. Then, the bag was numbered, rinsed with tap water, soaked for 50 min, dried at 65 °C to constant weight, and set aside.

To test the rumen’s capacity for degrading feed, three Hu sheep with fistulas were used. We started the experiment by placing a nylon bag into the sheep’s rumen abdominal sac. The bags were incubated for 0, 6, 12, 24, 36, and 48 h in the rumen (0 h was used to determine the soluble fraction), using different methods of simultaneously putting in and taking out. A total of 1.5 kg DM of grass-based 40% and 60% sheep-fattening pellets was provided to the Hu sheep. The bags were withdrawn from the rumen after incubation, washed under running water for 20 to 25 min until the effluent water was clear, and then dried for 48 h in a 65 °C oven. The contents of conventional nutrients in the original samples and those at different time points were measured, and the soluble fraction was estimated to correct the weight of feed samples and calculate the effective degradation.

### 2.7. Statistical Analysis

One-way analysis of variance (ANOVA) was used to analyze the data in the general linear model (GLM) function of the SAS software (version 9.4, SAS Institute, Cary, NC, USA). One-way analysis of variance was used to assess the data on chemical composition, fermentation quality, and anti-nutritional factors (ANOVA). A two-way ANOVA with a mixed model was also performed on the data for aerobic stability. Tukey’s multiple comparisons test was used to calculate the statistical difference between means, and the differences between means were deemed significant at *p* < 0.05. The exponential equation of Orskov and McDonald (1979), Y(t) = a + b (1 − e^−ct^), was used to determine the fraction of the incubated material that was degraded at time t (h of incubation); “a” stands for the water soluble and instantly degradable fraction, “b” stands for the potentially degradable fraction, and “c” stands for the fractional rate of degradation of fraction b (/h). The fractional rate of passage (k) was assumed to be 0.05/h for calculating the effective degradability (ED) of dry matter (DM) and starch, which was calculated as ED = a + (bc)/(c + k). The SAS NLIN procedure was used to calculate estimates for variables “a”, “b”, and “c” (version 9.4, SAS Institute, Cary, NC, USA).

## 3. Results

### 3.1. Quality of Ensiled TMR Fermentation

The fermentation quality and microbial composition of TMRs of anaerobic fermentation after 60 h of fermentation are shown in Table 2. A significant effect of replacing SBM with CSM and RSM on the pH value of TMR was observed (*p* < 0.05). The lactic acid and acetic acid content of the F-CSM group were significantly higher than those of the F-RSM group, respectively (*p* < 0.05). The propionic acid content was greater (*p* < 0.05) in the F-RSM groups than in the F-CSM. The replacement of RSM significantly decreased (*p* < 0.05) the NH_3_-N content of FTMR compared to that of the F-CSM group. The counts of lactic acid bacteria in the F-RSM and control groups did not differ significantly, whereas the counts of aerobic bacteria in the F-CSM group were significantly greater than those in the control and F-RSM groups (*p* < 0.05).

### 3.2. Chemical Composition of Ensiled Treatments

The chemical composition of the TMR and FTMR after 60 h is presented in Table 3. When comparing the groups, we noted that the DM content did not show a significant difference (*p* > 0.05). The OM content of the F-RSM group was significantly higher than that of the control and F-CSM groups (*p* < 0.05). The CP and ether extract (EE) contents of the F-CSM and F-RSM groups were significantly higher than those of the control group (*p* < 0.05). The NDF content was significantly decreased in the F-CSM and F-RSM groups, and the ADF content of the F-CSM group was lower than that of the control and F-RSM groups. The NDIP and ADIP contents of the F-RSM group were significantly higher than those of the other groups (*p* < 0.05).

### 3.3. Aerobic Stability of TMR 

Dynamic changes in temperature were monitored in non-fermented TMR and FTMR within seven days of air exposure (Figure 1). For non-fermented TMR, heating was observed in all treatments during air exposure. The temperature of the control group increased much more promptly than that of the F-CSM and F-RSM groups. Deterioration occurred in each group within 24 h and 48 h, respectively, indicated by the temperatures exceeding the ambient temperature by more than 2 °C. Subsequently, the temperature in all the groups decreased. Aerobic deterioration was accompanied by increasing pH values in both FTMRs. After 72 h of air exposure, the pH value of the TMRs increased to more than 7.0 in the control and F-RSM groups, but not more than 5.5 in the F-CSM group (*p* < 0.05).

The changes in the fermentative characteristics during aerobic exposure are shown in Table 4. With the extension of the aerobic-exposure time, the lactic acid concentration of all treatment groups significantly decreased (*p* < 0.05). The concentration of acetic acid was significantly reduced after 12 h in all the groups. In the control group, the acetic acid concentration was significantly lower than that in the F-CSM and F-RSM groups, whereas the concentration in the F-RSM group decreased at the beginning of air exposure (*p* < 0.05). The propionic acid concentration, like the acetic acid concentration, dropped quickly after exposure to air for 48 h. The F-CSM propionic acid group had slightly higher levels than the control and F-RSM groups (*p* < 0.05).

Changes in the composition of the microbial population in the FTMR were observed during the air-exposure period (Table 5). For FTMR, during the air-exposure time, the lactic acid bacteria (LAB) population initially increased, the F-RSM group was significantly higher than the lactic acid populations of the other two groups, and yeast counts were above 5.0 log_10_ CFU/g DM in all groups on the air-exposure day; after air exposure for 72 h, the yeast counts decreased and showed a significant difference among the groups (*p* < 0.05). The numbers of aerobic bacteria increased rapidly and were subsequently maintained at a level of approximately 8.3–8.7 log_10_ CFU/g DM for all treatments.

### 3.4. Anti-Nutritional Factors of Ensiled TMR

As presented in Figure 2a, the FG concentration decreased by half during fermentation and did not change after air exposure, and there was no significant difference among the exposure times (*p* > 0.05). Compared with non-fermented TMR, glucosinolate decreased after fermentation, remained unchanged during the entire air-exposure time, and had a greater than 50% detoxification rate (Figure 2b).

### 3.5. Ruminal Degradation Characteristics of Various TMR

Table 6, Table 7, Table 8 and Table 9 show the results of the changes in the ruminal disappearance and degradation characteristics of DM, CP, NDF, and ADF. The rumen disappearance of DM, CP, NDF, and ADF increased with the increasing rumen residence time; DM disappearance showed no significant difference between treatment groups at the same time. The rumen disappearance of CP at 24 h, 36 h, and 48 h for the control group was significantly higher than for the F-CSM and F-RSM groups (*p* < 0.05), but at 6 h and 12 h, there was no significant differences between each group. With the exception of the 12 h point, there was no difference among NDF and ADF in regard to rumen disappearance. The DM degradation parameters showed that the degradation rate of slow degradation part “c” was similar among treatments (*p* > 0.05) and that the rapid degradation part “a”, slow degradation part “b”, potential degradable part “a+b”, F-CSM, and F-RSM groups did not significantly differ from each other. The effective degradation rate (ED) was significantly higher in the F-RCM group compared to the F-CSM and control groups (*p* < 0.05). Although all CP degradation parameters for the F-CSM and F-RSM groups were significantly higher than those of the control group (*p* < 0.05), they did not significantly differ among themselves. NDF and ADF degradation parameters “a”, “a+b”, and “ED” in the F-CSM group were significantly greater than for the F-CSM and control groups (*p* < 0.05); slow degradation part “b” in the CSM group was higher than it was for the other two groups; and the “c” rate in the control group was higher than it was for the F-CSM and F-RSM groups. 

## 4. Discussion

### 4.1. Quality of Ensiled TMR Fermentation

TMR, a mixture of concentrate and roughage, is widely used as ruminant feed. FTMR is a method used to ferment feed under anaerobic conditions in a tightly sealed container, with the potential to enhance nutrient utilization and extend the shelf life of feed [28]. Numerous studies have shown that inoculating silage with bacteria can cause a decrease in the pH during fermentation [29]. In this study, a significant effect of fermented total mixed ration containing CSM and RSM replacing non-fermented TMR with soybean meal on the pH value of TMR was observed. Acetic acid has anti-fungal properties, which help reduce organism deterioration in ensiled mass and improve fermentation quality [30]. According to Kondo et al. [31], the proteolysis that occurs throughout fermentation causes an increase in NH_3_-N content. In their study, Driehuis et al. [32] found an increase in the NH_3_-N concentration in FTMR compared to non-fermented TMR; however, in our study, the NH_3_-N content of fermented TMR was decreased. The reason may be the influence of the anti-nutritional factors such as the CSM-free gossypol concentrate and RSM glucosinolate concentrate.

### 4.2. Chemical Composition of Ensiled TMR

Inoculated microbes increased the CP levels of both of the fermented TMRs with CSM/RSM [31]. When TMR was fermented, the CP content increased. Similarly, Kondo et al. [31] observed that TMR increased the CP content after ensiling, as compared to that before ensiling. However, our findings of lower NDF and ADF concentrations in all fermented TMR compared to non-fermented TMR might be explained [19] by the coexistence of the enzymolysis and acidolysis of cell-wall components during silage fermentation.

### 4.3. Aerobic Stability of Ensiled TMR

An increase in pH is an indicator of the aerobic deterioration of silage because lactic acid is consumed by yeasts during aerobic exposure, and the silage becomes favorable to the growth of other undesirable microorganisms, such as molds and bacteria [33]. The fermented TMR temperature of all groups increased significantly after 8 h of aerobic exposure, with the non-fermented control group experiencing the greatest increase. However, even after 48 h of aerobic exposure, the non-fermented control group temperature was more than 2 °C above the ambient temperature.

In a study by Ohyama et al. [34], remarkable decreases in lactic acid concentrations during air exposure occurred in most silages, primarily because lactic acid can be used as a substrate by some aerobic microorganisms, especially yeasts, in the presence of air. However, in the F-CSM group in our study, the lactic acid concentration significantly decreased during aerobic deterioration. A similar result was obtained in a previous study by Wang and Nishino [17]. This trend could be due to the combined activity of lactic-acid-assimilative microorganisms and facultative anaerobic lactic acid bacteria [17].

Acetic acid has been found to be one of the most effective substances for the inhibition of spoilage microorganisms and aerobic bacteria. In all groups, the acetic acid concentrations decreased significantly during deterioration, mainly due to volatilization under aerobic conditions. Although the presence of acetic acid bacteria was not determined in this study, in the study by Spoelstra et al. [35], several oxidation reactions of acetic acid by acetic acid bacteria may also occur during the extended phase of aerobic deterioration.

Bernardes et al. [36] reported that acid tolerance is a general characteristic of yeasts. Yeasts can oxidize fermentation products, leading to an increase in pH and the proliferation of aerobic microorganisms in air-exposed silage. Both Nishino et al. [37] and Kung et al. [38] reported that, when exposed to aerobic conditions, lactic acid is degraded by yeasts and aerobic microorganisms, resulting in aerobic deterioration. Filya et al. [39] thought that silages deteriorated when the yeast population exceeded 5.0 log_10_ CFU/g. For the FTMR in this study, the yeast population increased during air exposure; the control and F-CSM groups had higher numbers than F-RSM. After 72 h, the yeast significantly decreased in all groups, but the population was still greater than 5.0 log_10_ CFU/g. The non-fermented TMR was more prone to aerobic spoilage than the F-CSM and F-RSM. After air exposure, aerobic bacterial growth increased, with the control group outnumbering the other two groups.

### 4.4. Anti-Nutritional Factors of Ensiled TMR

CSM and RSM contain anti-nutritional factors that may disrupt nutrient availability, cause toxicity, and impair animal performance [40]. Our study’s FG degradation was lower than that reported by Tang et al. [41], who reduced FG in solid-state fermented cotton meal from 0.82 to 0.21 g/kg. Similarly, FG was reduced from 90.0 to 30.0 mg/kg in Xiong et al. [42]. Sun et al. [43] found that CSM with fermented *B. subtilis* BJ supplementation significantly reduced FG levels and increased CP levels. In this study, the FG concentration did not change significantly after the FG concentration decreased by half. The reduction in FG concentration may be the result of gossypol binding to microbial enzymes that break down gossypol during the fermentation of TMR with CSM [44]. In line with the earlier findings of Ahmed et al. [43], the current study revealed that the increased protein content by solid-state fermentation with *Lactobacillus salivarius* ranged from 41.2% to 42.2%, and the reduction in glucosinolates ranged from 22.0 to 13.6 mmol/g. Our results for glucosinolate degradation are consistent with those of previous studies, which decreased the glucosinolate concentration and detoxification rate by more than 50%. Pal et al. [45], Hu et al. [46], and Tripathi et al. [47] showed that the loss of glucosinolates leads to the creation of glucose and sulfur molecules by microbial enzymes during fermentation.

### 4.5. Ruminal Degradation Characteristics of Various TMR

The degradation of feed in the rumen is essentially a series of effects of rumen microbial physiological activities on feed nutrients. With the increase in the residence time of feed in the rumen, the rumen disappearance of feed increases gradually. Dietary CP disappearance is related to feed stock characteristics, such as anti-nutrition factors and the cell-wall barrier effect [48]. In this experiment, the content of the fast degradation part of the control group was lower than that of the F-CSM and F-RSM groups, while the content of the slow degradation part was higher than that of the two groups. However, the effective degradation rate of the control group was 34.46%, and that of the F-CSM and F-RSM groups was 46.93% and 45.38%, respectively. This was because the degradation rate of the slow degradation part in the two groups was 12.47% and 10.92% higher than that in the control group. NDF and ADF in feed mainly come from cell walls and are difficult to digest by ruminants [49]. The fermentation function of rumen microorganisms is related to NDF and ADF in feed. In this study, the rumen effective disappearance of NDF was mostly higher than that of ADF because the main components of ADF were lignin, cellulose, and silica. ADF is almost not decomposed and is not utilized in the rumen, so the disappearance of ADF in the rumen is generally low. According to the degradation dynamic parameters in this study, F-CSM and F-RSM were superior to the control group, indicating that fermentation was beneficial to improve the effective degradation rate and facilitate the digestion of NDF and ADF in the rumen.

## 5. Conclusions

In conclusion, the fermented TMR inoculations *B. clausii* and *S. cariocanus* affect the fermentation quality and nutrient composition. After fermentation, the anti-nutritional factors such as the FG concentration in F-CSM and F-RSM’s glucosinolate concentration decreased, the detoxification rate was more than 50%, and, thus, we achieved detoxification. In addition, the replacement of SBM by CSM and RSM in TMR by fermentation improved the nutrient degradation. Meanwhile, the effective disappearance of nutrients in the rumen was increased. 

## Figures and Tables

**Figure 1 animals-13-02730-f001:**
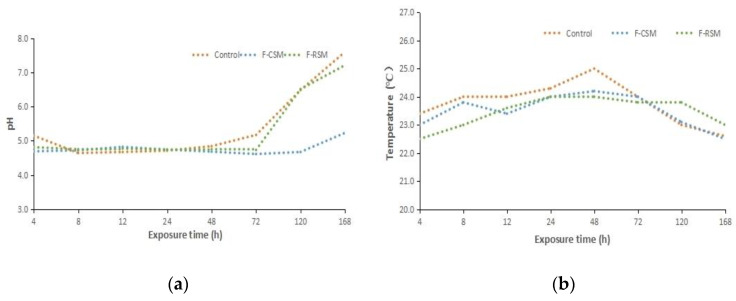
Dynamic changes in temperatures and pH during air exposure of FTMR: (**a**) temperature and (**b**) pH. Control, total mixed ration with soybean meal (non-fermented); F-CSM, fermented total mixed ration with cottonseed meal; F-RSM, fermented total mixed ration with rapeseed meal.

**Figure 2 animals-13-02730-f002:**
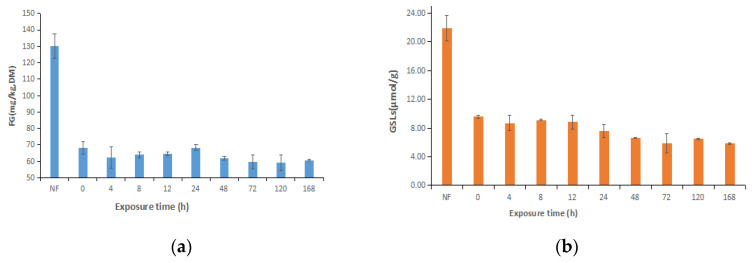
Dynamic changes in anti-nutritional contents during air exposure of F-CSM and F-RSM: (**a**) FG and (**b**) GSLs. FG, free gossypol; GSLs, glucosinolates; NF (**a**), non fermented total mixed ration with cottonseed meal; NF (**b**), non fermented total mixed ration with rapeseed meal.

**Table 1 animals-13-02730-t001:** Components and chemical composition of total mixed rations (DM basis).

	Groups ^1^		
Items	Control	F-CSM	F-RSM
Ingredients (%)			
Corn	34.00	33.55	33.48
Wheat bran	12.00	12.00	12.00
Soybean meal	10.00	0.00	0.00
Cottonseed meal	0.00	10.00	0.00
Rapeseed meal	0.00	0.00	10.00
Fat powder	0.00	0.30	0.30
Urea	0.00	0.15	0.22
Whole plant corn silage	20.00	20.00	20.00
Corn stalk	20.00	20.00	20.00
Premix ^2^	4.00	4.00	4.00
Total	100.00	100.00	100.00

^1^ Control, total mixed ration with soybean meal (non-fermented); F-CSM, fermented total mixed ration with cottonseed meal; F-RSM, fermented total mixed ration with rapeseed meal. ^2^ The premix provided the containing per kg of diets: vitamin A (IU) = 5000, vitamin D3 (IU) = 2000, vitamin E (IU) = 40, niacin (mg) = 300, biotin(mg) = 1, Fe (mg) = 7.5, Zn (mg) = 6, Mn (mg) = 6.2, I (mg) = 12.5, Co (mg) = 12.5, Se (mg) = 0.5, Ca (g) = 8, P (g) = 3, and NaCl (g) = 25.

**Table 2 animals-13-02730-t002:** The fermentative characteristics and microbial compositions of mixed-meal-replaced soymeal meal in TMRs (DM basis).

Fermented Parameters	Control	F-CSM	F-RSM	SEM	*p*-Value
pH	4.49 ^b^	4.58 ^ab^	4.78 ^a^	0.58	0.03
Lactic acid, g/kg, DM	-	128.82 ^a^	126.25 ^b^	0.93	0.02
Acetic acid, g/kg, DM	-	15.94 ^a^	15.18 ^b^	0.16	0.01
Propionic acid, g/kg, DM	-	0.11 ^b^	0.12 ^a^	0.01	<0.001
Butyric acid, g/kg, DM	-	-	-	-	
Ammonia nitrogen, g/kg	-	2.03 ^a^	1.37 ^b^	0.21	0.05
Microbial compositions					
Lactic acid bacteria, log_10_ CFU/g DM	7.77 ^a^	7.40 ^b^	7.62 ^ab^	0.07	0.03
Aerobic bacteria, log_10_ CFU/g DM	6.67 ^b^	8.06 ^a^	7.43 ^b^	0.12	0.01
Yeast, log_10_ CFU/g DM	7.47	7.22	7.18	0.07	0.21

Control, total mixed ration with soybean meal (non-fermented); F-CSM, fermented total mixed ration with cottonseed meal; F-RSM, fermented total mixed ration with rapeseed meal. SEM, standard error of the means. ^a,b^ *p* < 0.05. Note: For the statistical analysis, we compared fermented F-CSM and F-RSM with non-fermented CK.

**Table 3 animals-13-02730-t003:** The chemical composition of mixed-meal-replaced soymeal meal in TMRs (DM basis).

Items	Control	F-CSM	F-RSM	SEM	*p*-Value
DM (fresh basis)	55.85	56.86	56.9	0.12	0.21
OM	88.99 ^c^	90.58 ^b^	91.81 ^a^	0.52	<0.001
CP	12.14 ^c^	13.44 ^a^	13.15 ^b^	0.56	0.02
EE	2.06 ^b^	2.29 ^b^	3.31 ^a^	0.24	0.01
NDF	29.31 ^a^	26.51 ^b^	24.86 ^c^	0.83	0.01
ADF	17.72 ^a^	14.75 ^b^	17.72 ^a^	0.64	0.01
NDIP	3.22 ^b^	3.69 ^ab^	4.27 ^a^	0.20	0.05
ADIP	3.47 ^b^	3.65 ^b^	5.53 ^a^	0.43	0.02
TP	6.50	6.88	7.52	0.25	0.31
Starch	21.45	31.89	32.38	2.65	0.15
GE (MJ/kg DM)	20.10	21.10	21.23	0.89	0.12

Control, total mixed ration with soybean meal (non-fermented); F-CSM, fermented total mixed ration with cottonseed meal; F-RSM, fermented total mixed ration with rapeseed meal. DM, dry matter; OM, organic matter; CP, crude protein; EE, ether extraction; NDF, neutral detergent fiber; ADF, acid detergent fiber; NDIP, neutral insoluble protein; ADIP, acid insoluble protein; TP, true protein, GE, total energy. F-RSM groups. SEM, standard error of the means. ^a,b,c^ *p* < 0.05.

**Table 4 animals-13-02730-t004:** Dynamic changes in fermentation during air exposure of FTMR.

		Exposure Time (h)									*p* Value		
Item	Treatment	4	8	12	24	48	72	120	168	SEM	T	D	T × D
Lactic acid g/kg DM	Control	114.28 ^Ba^	108.54 ^Ba^	97.91 ^Bab^	80.90 ^Bc^	86.61 ^Bbc^	60.69 ^Cd^	11.81 ^Ce^	7.99 ^Cf^	3.17	<0.0001	<0.0001	<0.0001
	F-CSM	138.02 ^aAb^	146.15 ^Aa^	125.4 ^Ac^	115.26 ^Ac^	113.15 ^Ac^	103.87 ^Ad^	97.35 ^Ad^	87.65 ^Ae^				
	F-RSM	119.67 ^Ba^	94.38 ^Bb^	92.97 ^Cc^	87.57 ^Bc^	85.66 ^Bc^	80.22 ^Bc^	67.22 ^Bd^	67.69 ^Bd^				
Acetic acid g/kg DM	Control	14.23 ^Bb^	10.11 ^Ce^	14.61 ^Cb^	12.64 ^Cc^	16.03 ^Aa^	11.27 ^Bd^	4.53 ^Bf^	4.63 ^Bf^	0.360	<0.0001	<0.0001	<0.0001
	F-CSM	21.29 ^Aa^	17.34 ^Bd^	19.86 ^Ab^	18.42 ^Ac^	15.54 ^Be^	14.48 ^Af^	8.79 ^Ag^	7.15 ^Ah^				
	F-RSM	20.26 ^Aa^	19.66 ^Ab^	16.51 ^Bc^	16.67 ^Bc^	13.23 ^Cd^	10.84 ^Ce^	3.60 ^Cg^	3.80 ^Cf^				
Propionic acid g/kg DM	Control	0.23 ^Cb^	0.26 ^Cb^	0.38 ^Ba^	0.32 ^B^	0.28 ^Bab^	0.19 ^Bc^	0.15 ^Bd^	0.11 ^Be^	0.010	<0.0001	<0.0001	<0.0001
	F-CSM	0.51 ^Aa^	0.43 ^Ab^	0.43 ^Ab^	0.41 ^Ab^	0.37 ^Ac^	0.26 ^Ad^	0.26 ^Ad^	0.18 ^Ae^				
	F-RSM	0.47 ^Ba^	0.38 ^Bb^	0.36 ^Bb^	0.27 ^Cc^	0.25 ^Bc^	0.21 ^Bc^	0.24 ^Ac^	0.17 ^Ad^				
Ammonia nitrogen g/kg AN	Control	1.82 ^Be^	2.15 ^Ac^	1.52 ^Bf^	1.50 ^Bf^	1.94 ^Bd^	1.81 ^Ae^	3.52 ^Ab^	3.91 ^Aa^	0.029	<0.0001	<0.0001	<0.0001
	F-CSM	0.94 ^Cc^	0.32 ^Ce^	1.56 ^Bb^	1.54 ^Bb^	1.97 ^Ba^	0.81 ^Bd^	0.96 ^Cc^	0.85 ^Cd^				
	F-RSM	2.54 ^Ac^	1.93 ^Bg^	2.12 ^Ae^	1.79 ^Af^	3.09 ^Aa^	1.81 ^Af^	2.20 ^Bd^	2.84 ^Bb^				

Control, total mixed ration with soybean meal (non-fermented); F-CSM, fermented total mixed ration with cottonseed meal; F-RSM, fermented total mixed ration with rapeseed meal; SEM, standard error of the means; T, treatment; D, exposure time; ^a,b,c,d,e,f,g,h^ *p* < 0.05; ^A,B,C^ *p* < 0.01.

**Table 5 animals-13-02730-t005:** Dynamic changes in microbial composition characteristics during air exposure of FTMR.

		Explore Time (h)									*p* Value		
Item	Treatment	4	8	12	24	48	72	120	168	SEM	T	D	T × D
Lactic acid bacteria, log 10 cfu/g DM	Control	7.85 ^Ad^	8.08 ^bc^	8.39 ^Ab^	8.15 ^Bbc^	8.21 ^Bb^	8.26 ^Bb^	8.30 ^b^	8.55 ^Ba^	0.024	<0.0001	0.0153	<0.0001
	F-CSM	7.55 ^Be^	7.98 ^c^	7.83 ^Bd^	8.37 ^Aab^	8.47 ^Aab^	8.38 ^ABab^	8.32 ^b^	8.55 ^Ba^				
	F-RSM	7.96 ^Ac^	7.91 ^c^	8.16 ^Ab^	8.31 ^Ab^	8.46 ^Ab^	8.50 ^Ab^	8.46 ^b^	8.81 ^Aa^				
Aerobic bacteria, log10 cfu/g DM	Control	8.21 ^Ad^	8.26 ^Acd^	8.28 ^Ac^	8.21 ^Ad^	8.25 ^Acd^	8.21 ^Ad^	8.47 ^Ab^	8.77 ^Aa^	0.034	<0.0001	0.0153	<0.0001
	F-CSM	7.75 ^Bd^	7.78 ^Bcd^	7.58 ^Ce^	7.68 ^Be^	7.67 ^BCe^	7.80 ^Cc^	8.04 ^BCb^	8.33 ^BCa^				
	F-RSM	7.58 ^Cf^	7.72 ^Be^	7.96 ^Bb^	7.21 ^Cg^	8.14 ^ABc^	8.06 ^Bd^	8.35 ^ABb^	8.83 ^ABa^				
Yeast, log10 cfu/g DM	Control	6.42 ^Bg^	6.96 ^Af^	7.95 ^Ad^	8.51 ^Ac^	8.95 ^Ab^	9.12 ^Aa^	7.78 ^Ae^	6.45 ^Ag^	0.015	<0.0001	<0.0001	<0.0001
	F-CSM	6.52 ^Af^	6.62 ^Be^	7.80 ^Bc^	8.00 ^Bb^	8.07 ^Cb^	9.05 ^Ba^	7.11 ^Cd^	5.09 ^Cg^				
	F-RSM	6.25 ^Ce^	6.40 ^Cd^	7.46 ^Cc^	7.45 ^Cc^	8.47 ^Bb^	8.49 ^Ca^	7.48 ^Bc^	6.01 ^Bf^				

Control, total mixed ration with soybean meal (non-fermented); F-CSM, fermented total mixed ration with cottonseed meal; F-RSM, fermented total mixed ration with rapeseed meal; SEM, standard error of the mean. T, treatment; D, exposure time; ^a,b,c,d,e,f,g^ *p* < 0.05; ^A,B,C^ *p* < 0.01.

**Table 6 animals-13-02730-t006:** Dynamic changes in the dry-matter-degradation characteristics of various TMRs.

Item	Control	F-CSM	F-RSM	SEM	*p*-Value
DM rumen disappearance (%)					
6 h	19.45	22.66	22.48	1.19	0.48
12 h	26.78	29.09	27.28	1.17	0.72
24 h	31.26	36.12	33.51	1.58	0.48
36 h	34.80	41.94	39.90	1.61	0.18
48 h	38.54	44.49	42.13	1.70	0.37
DM rumen-degradation parameters					
a (%)	13.81 ^a^	15.44 ^ab^	16.97 ^b^	0.54	0.02
b (%)	26.97 ^a^	33.79 ^b^	33.18 ^b^	1.13	<0.001
c (%.h^−1^)	0.05	0.04	0.03	0.01	0.13
a+b (%)	40.78 ^a^	49.23 ^b^	50.15 ^b^	1.52	<0.001
ED (%)	28.40 ^a^	35.57 ^b^	38.96 ^c^	0.58	<0.001

a, rapid degradation part (%); b, rapid degradation part (%); c, rapid degradation part (% h^−1^); ED, effective degradation rate (%). In the same row, values with different lowercase-letter superscripts indicate a significant difference (*p* < 0.05). The same as below.

**Table 7 animals-13-02730-t007:** Dynamic changes in the crude-protein-degradation characteristics of various TMRs.

Item	Control	F-CSM	F-RSM	SEM	*p*-Value
CP rumen disappearance (%)					
6 h	35.10	31.74	30.19	1.09	0.18
12 h	46.07	42.80	41.71	0.98	0.17
24 h	58.03 ^a^	44.28 ^b^	43.55 ^b^	1.79	<0.001
36 h	61.94 ^a^	52.51 ^b^	49.30 ^b^	1.56	<0.001
48 h	62.49 ^a^	55.80 ^ab^	51.91 ^b^	1.34	0.03
CP rumen-degradation parameters					
a (%)	15.30 ^a^	26.42 ^b^	25.34 ^b^	1.79	<0.001
b (%)	48.39 ^a^	34.41 ^b^	33.63 ^b^	2.41	<0.001
c (%.h^−1^)	0.09 ^a^	0.04 ^b^	0.04 ^b^	1.00	<0.001
a+b (%)	63.69 ^a^	60.83 ^b^	58.97 ^b^	0.74	<0.001
ED (%)	34.46 ^a^	46.93 ^b^	45.38 ^b^	1.98	<0.001

a, rapid degradation part (%); b, rapid degradation part (%); c, rapid degradation part (% h^−1^); ED, effective degradation rate (%). In the same row, values with different lowercase-letter superscripts indicate a significant difference (*p* < 0.05). The same as below.

**Table 8 animals-13-02730-t008:** Dynamic changes in the NDF-degradation characteristics of various TMRs.

Item	Control	F-CSM	F-RSM	SEM	*p* Value
NDF rumen disappearance (%)					
6 h	30.66	26.45	27.59	1.13	0.30
12 h	44.70 ^a^	36.16 ^ab^	37.34 ^b^	1.16	0.02
24 h	51.68	51.69	49.51	1.13	0.69
36 h	57.21	56.50	56.96	1.05	0.96
48 h	58.53	59.32	59.18	1.13	0.95
NDF rumen-degradation parameters					
a (%)	9.82 ^b^	6.89 ^c^	12.95 ^a^	0.92	<0.001
b (%)	48.53 ^b^	53.45 ^a^	49.88 ^b^	0.79	<0.001
c (%.h^−1^)	0.10 ^a^	0.08 ^ab^	0.06 ^b^	0.01	0.01
a+b (%)	58.35 ^a^	60.34 ^a^	62.83 ^b^	0.71	0.01
ED (%)	27.82 ^b^	29.57 ^b^	37.68 ^a^	1.54	<0.001

a, rapid degradation part (%); b, rapid degradation part (%); c, rapid degradation part (% h^−1^); ED, effective degradation rate (%). In the same row, values with different lowercase-letter superscripts indicate a significant difference (*p* < 0.05). The same as below.

**Table 9 animals-13-02730-t009:** Dynamic changes in the ADF-degradation characteristics of various TMR.

Item	Control	F-CSM	F-RSM	SEM	*p* Value
ADF rumen disappearance (%)					
6 h	310.76	29.70	26.98	1.10	0.37
12 h	44.88 ^ab^	47.38 ^a^	39.46 ^b^	1.15	0.01
24 h	51.19	50.80	53.35	1.13	0.63
36 h	55.15	59.24	57.95	1.09	0.31
48 h	57.03	59.45	61.01	1.19	0.40
ADF rumen-degradation parameters					
a (%)	6.41	6.69	7.24	0.31	0.611
b (%)	49.68 ^c^	52.17 ^b^	55.10 ^a^	0.83	<0.001
c (%.h^−1^)	0.12 ^a^	0.11 ^a^	0.06 ^b^	0.01	<0.001
a+b (%)	56.09 ^c^	58.86 ^b^	62.34 ^a^	0.95	<0.001
ED (%)	22.79 ^b^	24.90 ^b^	35.56 ^a^	1.84	<0.001

a, rapid degradation part (%); b, rapid degradation part (%); c, rapid degradation part (% h^−1^); ED, effective degradation rate (%). In the same row, values with different lowercase-letter superscripts indicate a significant difference (*p* < 0.05). The same as below.

## Data Availability

The data that support the findings of this study are available upon request from the corresponding author.

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
