# Peer review of "Evaluating Fermentation Quality, Aerobic Stability, and Rumen-Degradation (In Situ) Characteristics of Various Protein-Based Total Mixed Rations"

_animals, 2023, doi:10.3390/ani13172730_

Round 1
Reviewer 1 Report (Previous Reviewer 1)
The experiment design of this paper is fatal because many factors will affect the indexes rather than one factor, and therefore I suggest rejecting it.
Author Response
Great thanks to you providing us nice comments as well as revision suggestions.
We would like to express our appreciation to you for carefully reviewing our paper and providing useful comments and suggestions.
Attach file is our response letter, please check it.
Great thanks again
Yan Tu

Reviewer 2 Report (Previous Reviewer 2)
Look for comments in the attached file!

I recommend some English professional editing! Most if it is understandable, but English can be improved!
Author Response
Dear Reviewer:
Great thanks to you providing us nice comments as well as revision suggestions.
We would like to express our appreciation to you for carefully reviewing our paper and providing useful comments and suggestions.
We have made revisions to the revised manuscript in response to the comments and
recommendations received. We have copied each reviewer's comment below and followed up with a comprehensive answer Additionally, we have used red colour in the revised portions of the manuscript by using “Track Changes” function.
Hopefully the revised version could be acceptable, and if there are any improper revision, please don’t hesitate to inform us.
Great thanks again,
Tu yan

Reviewer 3 Report (New Reviewer)
Proteins are an indispensable nutrient source for animal growth and development. The quality of protein feed affects the health and productivity of animals. Soybean meal is a high-quality protein feed, but its high price has seriously affected its use in the domestic animal breeding industry, especially in China. The purpose of this study was to evaluate changes in fermentation quality, chemical composition, aerobic stability, anti-nutritional factors and in situ disappearance characteristics of various protein-based total mixed rations. Soybean meal (control, non-fermented), fermented cottonseed meal (F-CSM ), and fermented rapeseed meal (F-RSM) group were used to prepare the TMR 30 with corn, whole-plant corn silage, corn stalks, wheat bran and premix. The test groups were inoculated at 50 % moisture with Bacillus clausii and Saccharomyces cariocanus and stored aerobically for 60 h. The nylon bag method was used to measure and study the rumen's nutrient degradation. In conclusion, the authors found that fermented TMR inoculations B. clausii and S. cariocanus improved the fermentation quality and nutrient composition, decreased the anti-nutritional factor content and thus achieved detoxification. Meanwhile, the effective disappearance of nutrients in the rumen was increased.
The work is well done but I have some remarks:
- The authors should improve the quality and resolution of figures: 1 (a), 1(b), 2 (a), 2(b),
- The authors should extend the conclusion part, it is very short
Minor editing of English language required
Author Response
Dear Reviewer:
Great thanks to you providing us nice comments as well as revision suggestions.
We would like to express our appreciation to you for carefully reviewing our paper and providing useful comments and suggestions.
We have made revisions to the revised manuscript in response to the comments and recommendations received. We have copied each reviewer's comment below and followed up with a comprehensive answer Additionally, we have used red colour in the revised portions of the manuscript by using “Track Changes” function.
Hopefully the revised version could be acceptable, and if there are any improper revision, please don’t hesitate to inform us.
Great thanks again,
Tu yan

Reviewer 4 Report (New Reviewer)
This manuscript gives the set of interesting results of nutritional and chemical quality of various fermented total mixed rations (TMRs) with the inclusion of different protein sources. The layout of the paper is sufficiently good. The English is also sufficiently good. The methodology is clear and detailed but requires some minor additional explanations. The results are presented in manner so that potential readers can clearly follow the given discussion. The presented results are not groundbreaking but great advantage of this manuscript is that authors presented the results of in situ TMRs disappearance in cannulated sheeps which gives much clearer overview of potential of investigated novel feeds in practical feeding. Therefore, this manuscript would be interesting for professionals and not just academia. My comments on presented work are as follows:
Line 16: Please put the capital letter at the beginning of the sentence.
Line 18: Please give full terms before using abbreviation in this section of the manuscript. Although the abbreviations are explained in the introduction section, please put it here also so that readers can have full info by just reading simple summary of wok.
Line 21: Please put Latin names in italic.
Lines 65-66: Please rephrase this sentence. The rapeseed is crushed in order to produce oil, meal is just a byproduct of oil extraction.
Line 73: Please remove the word “methods”.
Line 73-74: Please put Latin names in italic.
Lines 144-146: Please correct the English in this sentence. It hinders fluent reading as it is now.
Lines 164-165: Which device was used for blending FTMR sample with water? Please specify it.
Line 169: Please use passive tense and try to avoid use of personal pronouns. This is valid for the whole text.
Line 177-178: Please provide short explanation what is determined and quantified after centrifugation. The mixture? What in it? What was measured by colometric method? Yes, the reference is given in text, but please give more details in text.
Line 193: Define DM and OM in text in this whole section and then use abbreviations in rest of the manuscript.
Line 306: Provide Figure 1 in higher quality.
Line 312: Explanation of abbreviations in Table 4 needs to be corrected, CK is not fully defined.
Line 413: Delete “our result”.
Lines 471-474: This part of text fits more into Introduction section. Remove it.
The authors need to rephrase some parts of text but the changes in English are not required at great extent.
Author Response
Dear Reviewer:
Great thanks to you providing us nice comments as well as revision suggestions.
We would like to express our appreciation to you for carefully reviewing our paper and providing useful comments and suggestions.
We have made revisions to the revised manuscript in response to the comments and recommendations received. We have copied each reviewer's comment below and followed up with a comprehensive answer Additionally, we have used red colour in the revised portions of the manuscript by using “Track Changes” function.
Hopefully the revised version could be acceptable, and if there are any improper revision, please don’t hesitate to inform us.
Great thanks again,
Tu yan

This manuscript is a resubmission of an earlier submission. The following is a list of the peer review reports and author responses from that submission.
Round 1
Reviewer 1 Report
This manuscript explored the fermentation quality, aerobic stability, and rumen degradation (in situ) characteristics of various protein-based total mixed rations, which is interesting and falls into the scope of the journal Animals. However, the experiment design of this manuscript is greatly inappropriate. First, this manuscript is not a real "One-way", a few factors can affect the indexes. For instance, a mixture of microbial additives was used in F-CSM and F-RSM, rather than Control. Second, fermentation also will affect the indexes, but the control is not a fermentation group. In addition, the fermentation conditions for F-CSM and F-RSM were not in accordance (60 h for 32℃ and 28 ℃, respectively). More details can be available as follows.
L16, whole-plant corn silage
L17, "Bacillus clausii" and "Saccharomyces cariocanus" should be italic, also in L 467, 496, 524 and 533
L18, please keep in accordance in the text between "h" and "hours" (L18 and 130), I suggesting using "h"
L31, the abbreviations can be used only on the condition that the full words appears at least three time in the text. Therefore, there no need to use the abbreviations for DM, CP, NDF and ADF
L48, a CP content of
L212, "p"
Tables 4 and 5, since the interaction was completely significant, the authors should slice the main factors
Reviewer 2 Report
The study is of interest and have merit, but it is immature and efforts mut be put into improvements before it can be published. Parts of material and methods must be better described. You appears to have been in hurry and have not taken time to do all parts properly. The same goes with presentation of results, especially the in sacco part which is not good, and I think wrongly estimated. I cannot redo your ED estimates!
I have not evaluated the discussion as I think you have to redo parts of that when revising. I will do that when/if I get the revised manuscript for review. I have stated reject in its present form, but you can improve and resubmitt for new evaluation if editor invite you after major revision.
Please look into the attached file for my comments and suggestions! Although I am harsh in my comments, parts are good so this is doable!
